

# An aerosol classification scheme for global simulations using the K-means machine learning method

Jingmin Li[1], Johannes Hendricks[1], Mattia Righi[1], Christof G. Beer[1]

[1]Deutsches Zentrum für Luft- und Raumfahrt (DLR), Institut für Physik der Atmosphäre, Oberpfaffenhofen, Germany

*Correspondence to*: Jingmin Li (Jingmin.Li@dlr.de)

**Abstract.** A machine learning K-means algorithm is applied to data of seven aerosol properties from a global aerosol simulation using EMAC-MADE3. The aim is to partition the aerosol properties across the global atmosphere in specific aerosol regimes. K-means is an unsupervised machine learning method with the advantage that an a priori definition of the aerosol classes is not required. Using K-means, we are able to quantitatively define global aerosol regimes, so-called aerosol clusters,

and explain their internal properties as well as their location and extension. This analysis shows that aerosol regimes in the lower troposphere are strongly influenced by emissions. Key drivers of the clusters' internal properties and spatial distribution are, for instance, pollutants from biomass burning/biogenic sources, mineral dust, anthropogenic pollution, as well as their mixing. Several continental clusters propagate into oceanic regions. The identified oceanic regimes show a higher degree of pollution in the northern hemisphere than over the southern oceans. With increasing altitude, the aerosol regimes propagate

from emission-induced clusters in the lower troposphere to roughly zonally distributed regimes in the middle troposphere and in the tropopause region. Notably, three polluted clusters identified over Africa, India and eastern China, cover the whole atmospheric column from the lower troposphere to the tropopause region. A markedly wide application potential of the classification procedure is identified and further aerosol studies are proposed which could benefit from this classification.

## 1 Introduction

Aerosols play an important role in the climate system (Boucher et al., 2013). They influence climate directly by scattering and absorption of solar and terrestrial radiation, as well as indirectly by modifications of cloud properties and their resulting radiative effects. The major components of atmospheric aerosols are mineral dust, black carbon (BC) and organic carbon, sulphate, nitrate, ammonium and sea salt. Due to their relatively short residence times, the contributions of these components, their state of mixing as well as the particle size distribution show a large spatial and temporal variability on the global scale

(e.g., Lauer and Hendricks, 2006; Li et al. 2009; Mann et al., 2010, 2014; Pringle et al., 2010; Aquila et al., 2011, Sessions et al., 2015, Kaiser et al., 2019). Additionally, their effects on clouds and radiation are highly variable due to the strong dependencies on the physical and chemical properties of the aerosols. This in combination with uncertainties in the current knowledge of key aerosol-related processes makes the quantification of aerosol-climate effects a challenge and results in





comparatively large uncertainties in the existing quantifications of the climate impact of anthropogenic aerosols (e.g., Boucher
et al. 2013; Myhre et al. 2017, Bellouin et al., 2020).

Global aerosol-climate models equipped with detailed representations of aerosol microphysical and chemical processes are essential tools for the quantification of aerosol-climate effects (e.g., Boucher et al. 1998; Takemura et al. 2005; Stier et al. 2005, 2006; Lauer et al. 2007; Hoose et al. 2008; Righi et al. 2013; Randles et al. 2013; Kipling et al. 2016; Myhre et al. 2017;
Bellouin et al., 2020; Righi et al. 2020). During the last decades, considerable attempts have been made by the global aerosol modelling community to develop improved descriptions of aerosol-climate interactions (e.g., Whitby et al. 1997; Ghan and Schwartz, 2007; Boucher et al., 2013, Riemer et al., 2019). Early modelling approaches considered only the mass of aerosol species. However, observations imply that the number, size distribution, and mixing state of aerosols are also critical factors for an accurate representation of aerosol-climate interactions (Albrecht et al. 1989). First attempts of representing the aerosol
size distribution and mixing state in global models started by the end of the 20$^{th}$ century (e.g., Whitby et al. 1997; Jacobson 2001). Due to limited computing capacities and the huge computational expenses of global aerosol-climate models, cost effective algorithms have been applied, for instance, lognormal representations of the aerosol size distribution (e.g., Stier et al. 2005; Lauer et al. 2005; Aquila et al. 2011; von Salzen 2006; Pringle et al., 2010; Kaiser et al. 2019). Such approaches allow for tracking soluble and insoluble aerosol particle components as well as their mixtures and facilitate the simulation of particle
number, mass concentration and size distribution. Beyond the direct radiative impact of aerosols, aerosol-cloud interactions are key processes driving the aerosol climate effects. Hence, parametrizations of aerosol activation in liquid clouds have been established (see Gahn et al., 2011, for a review). In addition, aerosol-induced formation of ice crystals attracts increasing attention (Kanji et al., 2017; Heymsfield et al. 2017). To represent the manifold ice formation pathways induced by a large number of different aerosol types in global aerosol-climate models, the applied microphysical cloud schemes as well as the
underlying aerosol sub-models have been further extended (e.g., Lohmann and Kärcher, 2002; Kärcher et al., 2006; Lohmann et al., 2007; Lohmann and Hoose, 2009; Hendricks et al., 2011; Kuebbeler et al., 2014; Righi et al., 2020).

The above examples demonstrate the growing complexity of global aerosol models which, consequently, results in a large number of parameters which describe the aerosol number concentration, size distribution and composition in global models
and makes the analysis, evaluation and interpretation of the model results a challenge. This is further complicated by the large spatial and temporal variability of the aerosol properties. Under these circumstances, analysing all relevant variables from a typical global model simulation can become unfeasible. New analysis methods are therefore required to gather information from the huge set of variables and their temporal and spatial variability. A powerful tool to facilitate the analysis of global aerosol model results is the partitioning of the model-simulated aerosol into different groups/clusters, each characterized by
specific properties. In the following, these groups will be called aerosol regimes. Information on how these aerosol regimes are distributed in space could be very helpful to obtain a concise but comprehensive view on the complex system of modelled aerosol parameters. Detailed knowledge of the spatial distribution of individual aerosol regimes could be the basis for further





analyses. For instance, observations within a specific aerosol regime can be combined for evaluating simulation results with regard to this specific aerosol type. Furthermore, model evaluation results based on observations limited in space and time

(e.g. aircraft-based field campaigns), could be generalized to a whole aerosol regime covering much larger areas and time periods, assuming that the systematic model biases to be corrected occur nearly homogenously throughout the whole cluster. In addition, knowledge of the properties and spatial extension of aerosol regimes could serve as supportive information for satellite retrieval and for the planning of further field campaigns for aerosol observation.

Previous aerosol classifications have been mainly conducted in the context of observational studies using measurements of aerosol microphysical and optical properties. For example, Groß et al. (2013, 2015) applied classification schemes to identify specific aerosol types and their mixtures based on lidar measurements and satellite data. Their classification procedure follows a tree structure where different aerosol microphysical and optical properties imply different classification branches. This allows to identify complicated vertical stratifications of different aerosol types throughout the atmosphere. Bibi et al. (2016) applied

multiple clustering techniques to analyse seasonal differences in prevailing aerosol types at four locations in India. Their classification is based on the analysis of pairs of aerosol optical properties gained from the Aerosol Robotic Network (AERONET) sun photometer measurements. Schmeisser et al. (2017) applied a similar multiple clustering technique to classify aerosol types based on surface-based observations of spectral aerosol optical properties from a global station network. Nicolae et al. (2018) classified six aerosol types using an artificial neural network applied to lidar measurement. The neural network

was trained with predefined data from different aerosol types. Applying similar algorithms to global model results using optical aerosol properties to classify aerosol types, however, could be problematic since the optical properties are derived quantities, which are calculated from primary (prognostic) quantities such as aerosol number, size and composition. These calculations also require additional assumptions, usually retrieved from measurements of, e.g. aerosol refractive indices, possibly implying increased uncertainties (Dietmüller et al. 2016). Hence new algorithms for aerosol classification based on primary aerosol

model parameters would be more appropriate.

In this study, we adopt the K-means machine learning clustering algorithm (Hartigan and Wong 1979) as a basis for identifying clusters of specific aerosol types in global aerosol simulations. This method partitions $n$ samples into $k$ clusters in which each sample is assigned to the cluster with the nearest distance to the clusters' centre (or cluster centroid). K-means belongs to the

class of unsupervised machine learning algorithms. This is especially useful when the classification criteria are unknown, as in the case of aerosol classification where the specific aerosol characteristics for the predominant regimes are not known a priori. K-means has already been applied in atmospheric research. For instance, it has been successfully used to distinguish clouds and aerosols in CALIOP/CALIPSO observations (Zeng et al. 2019). The present study aims to answer the following questions: (1) how can major aerosol regimes be identified in global aerosol simulations? (2) what is the spatial distribution of

these regimes? and (3) which aerosol types are dominant in which parts of the world? The K-means method is applied here to identify clusters of different aerosol types in global simulations. The spatial extension of these clusters is quantified. The





aerosol properties considered for the clustering process were simulated using the global chemistry-climate model system EMAC (the ECHAM/MESSy Atmospheric Chemistry, Jöckel et al. 2010, 2016) equipped with the aerosol microphysical sub module MADE3 (Modal Aerosol Dynamics model for Europe adapted for global applications, third generation, Kaiser et al.

2014, 2019). These analysed aerosol properties include the mass concentrations of mineral dust, BC, particulate organic matter (POM), sea salt, the sum of aerosol sulphate, nitrate and ammonium (SNA), as well as particle number concentrations in different aerosol size modes. The clustering analysis is conducted separately for the lower troposphere, the mid troposphere and the tropopause region.

The paper is structured as follows: Section 2 describes the model data and the analysis methods in detail. The results of the global clustering procedure are presented in Sect. 3, including separate discussions of the three predefined atmospheric layers. Further discussions about the limitation of the applied method and its potential applications are subject of Sect. 4. A summary of the main conclusions as well as an outlook are given in Sect. 5.

## 2 Data and methods

### 2.1 Model description and configuration

As a basis for aerosol classification in the present study, we analysed one of the global model simulations of Beer et al. (2020) performed with the global aerosol model EMAC-MADE3. MADE3 simulates nine different aerosol species (sulphate, ammonium, nitrate, the sea salt species sodium and chloride, BC, POM, mineral dust and aerosol water). These nine aerosols

species occur as three different internal mixtures (purely soluble particles, mixed particles consisting of an insoluble core with a soluble coating, and particles mainly composed of insoluble material and only very thin soluble coatings) within three size modes (Aitken-, accumulation- and coarse mode). This results in a total of nine aerosol modes. The model considers particle transformations due to coagulation and condensation, gas-particle partitioning and new particle formation. MADE3 was evaluated in detail in past studies and generally showed a good model performance. Kaiser et al (2014) demonstrated the ability

of MADE3 to represent aerosol microphysics processes such as new particle formation, gas-aerosol partitioning, and aerosol coagulation. Kaiser et al. (2019) further demonstrated the good agreement of MADE3 simulated BC, POM, gaseous species and particle number concentrations with various observations. Beer et al. (2020) extended the model setup of Kaiser et al. (2019) by including an online parametrization for wind-driven dust emissions (Tegen et al., 2002). The authors performed five model experiments for the time period 2000-2013 applying different horizontal and vertical model resolutions. The model

results were evaluated by comparison against observational data from the AERONET station network (Holben et al. 1998, 2001) and aircraft-based measurements from the SALTRACE field campaign (Weinzierl et al. 2017). The comparison in Beer et al. (2020) showed that a specific configuration (T63L31Tegen) outperforms the others thanks to its higher resolution and



the more detailed representation of dust emission processes. Hence, data from this simulation are selected for the clustering analysis in the present study.


For the chosen simulation Beer et al. (2020) applied EMAC in nudged mode, that is, model dynamics were constrained using ECMWF reanalysis data (Dee et al. 2011) including wind divergence and vorticity, temperature, and logarithm of the surface pressure for the years 1999 to 2013. 1999 was the spin-up year and is therefore not included in the analysis. Transient emission data for anthropogenic sources were used to match this simulation period. Anthropogenic emissions were chosen according to

the ACCMIP (Atmospheric Chemistry and Climate Model Intercomparison Project; Lamarque et al. 2010) inventory until the year 2000 and the RCP 8.5 scenario (Riahi et al. 2007, 2011) afterwards. Biomass burning emissions were taken from the Global Fire Emission Database version 4 (van der Werf et al. 2017). The wind-driven emissions of mineral dust were calculated online for every model time step following the dust parametrization developed by Tegen et al. (2002). As mentioned above, the model was applied at a T63L31 resolution, corresponding to a $1.9° \times 1.9°$ horizontal resolution and 31 vertical hybrid

pressure levels covering the vertical range from the surface up to 10 hPa. For a more detailed description of the simulation setup, we refer to Beer et al. (2020).

## 2.2 Data

Seven aerosol parameters extracted from the Beer et al. (2020) simulation are considered for the clustering process: the mass concentrations of mineral dust, BC, POM, sea salt, the sum of the sulphate, nitrate, and ammonium concentration (SNA), as

well as Aitken and Accumulation mode particle number concentration $N_{akn}$ and $N_{acc}$ of the combined aerosol species. Using number properties in addition to mass properties is helpful since the number ratio of small to large particles can change even when the total mass stays constant. The number concentrations of coarse mode particles are not taken into account to avoid duplicate information, since they are strongly correlated with the mass concentration of sea salt and mineral dust, owing to a comparatively small variability in the size distributions of the modelled mineral dust and sea salt particles. Since the size

distributions of the modelled Aitken and accumulation modes are more variable, the number concentrations of these particles are considered in addition to the corresponding mass concentrations. The clustering process is intended to identify grid points with similar climatological mean aerosol parameters, as a basis to classify the global aerosol distribution in different aerosol regimes.

The simulation data from years 2000 to 2013 are first reduced to multi-year (14 years) means to investigate the distribution of climatological aerosol regimes. To account for the vertical variability of aerosol properties, the model data at 31 vertical levels in the terrain following hybrid sigma pressure level are integrated to column values for three atmospheric layers. More specifically, we integrate model level L31-22 for the lower troposphere (up to ~700 hPa), L21-14 for the middle troposphere (~700 to ~300 hPa) and L13-6 for the tropopause region (~300 to ~100 hPa). Note that EMAC vertical levels are ordered top-

to-bottom. Due to the terrain following hybrid sigma pressure level concept, these layers only approximately correspond to





specific pressure levels. Deviations can occur in particular over elevated terrain (e.g., the Tibetan Plateau) where the pressure is lower in the layer than in other areas. This layer definition in the statistical analysis, however, is more flexible and can easily be adopted to the respective applications. We have chosen the specific definition described above just for a first demonstration of the clustering algorithm.

## 2.3 Method

The K-means algorithm used in this study is an unsupervised machine learning algorithm which does not require training data based on known and established classifications. It was first introduced by MacQueen (1967) and a more efficient version of K-means was developed by Hartigan and Wong (1979). K-means is a procedure based on the calculation of the squared Euclidean distance (Spencer, 2013). The Euclidean distance describes the distance between two points in the Euclidean space which can be spanned in any integer dimensions. Assuming that $p$ and $q$ are two points in a $j$-dimensional space, the Euclidean distance $d(p, q)$ between $p$ and $q$ is calculated by:

$$d(p,q) = \sqrt{(p_1 - q_1)^2 + (p_2 - q_2)^2 + \cdots + (p_j - q_j)^2} \quad (1)$$

The K-means method partitions a sample set into a predefined number of clusters ($k$) using minimization within cluster variances. The basic input of the algorithm is a sample $X= \{x^1, \ldots, x^n\}$ with $x^m = (x_1{}^m, x_2{}^m, \ldots x_j{}^m)$ and $m \in \{1, \ldots, n\}$, where $n$ is the number of data points and $j$ is the number of variable properties. The sample $X$ is grouped into $k$ cluster subsets ($S_1, S_2, \ldots S_k$) by minimizing the sum of the variances within each cluster $S_{i=1, \ldots, k}$ as follows:

$$\underset{S}{\arg\min} \sum_{i=1}^{k} \sum_{X \in S_i} \|x - \mu_i\|^2 \qquad (2)$$

where $\mu_i$ is the center of cluster $S_i$ (also called cluster centroid) and the term $\|x - \mu_i\|$ is a simplified notation of equation (1) describing Euclidean distances between all samples in $x$ and their cluster center $\mu_{i=1}^{k}$ in $j$ Euclidean dimensions. The *argmin* operator identifies the set of clusters $S_{i=1, \ldots, k}$ which minimizes the total sum of the Euclidean distance. By applying this procedure, each member of $X$ is assigned to a specific cluster. K-means is a stepwise forward iteration process. In the first step, the cluster centroids are assigned randomly and a prototype of the clusters is first estimated using equation (2). Then, in the second step, the cluster centroids are replaced by prototype cluster means. These two steps are iterated until the cluster centroids change only marginally or even stay constant. At this point the corresponding clusters can be regarded as the optimal set of clusters.

Choosing the appropriate $k$ for K-means algorithm is not straightforward. It requires a combination of clustering evaluation metrics in combination with expert judgement to evaluate the plausibility of the obtained clusters. Two clustering evaluation metrics commonly used are the sum of squared errors (SSE) and the silhouette coefficient (SC; Rousseeuw, 1987). The SSE is the sum of squared errors calculated between all data points and their cluster centre:

$$SSE = \sum_{i=1}^{k} \sum (X - \mu_i)^2 \quad (3)$$





By plotting the SSE as a function of $k$ and looking for the elbow point on the resulting curve, it is possible to identify the level of a mathematical optimization beyond which the further decrease in the error with increasing $k$ is no longer worth the additional computing cost.

The SC is a metric to validate the consistency/similarity within data of clusters and is defined as:

$$SC = \frac{\sum_{i=0}^{n} sc(i)}{n}, (4)$$

Where $sc(i) = \frac{b(i) - a(i)}{\max\{a(i), b(i)\}}$ (5)

where $a(i)$ is the averaged distance of sample $i$ to all other samples within a cluster and $b(i)$ is the averaged distance of sample $i$ to all samples of its nearest cluster that the sample i is not a part of. SC values range from $-1$ to $+1$, with a higher value

indicating that samples are well matched to the cluster they were assigned to, while they fit poorly to other clusters (Rousseeuw, 1987).

In this study, we apply the K-means clustering algorithm and calculate cluster evaluation metrics using the Python machine learning package scikit-learn (Pedregosa et al. 2011). The individual model grid points of the global simulation ($192 \times 96 = 18432$

points at the chosen T63 horizontal resolution) are assigned to $k$ clusters based on the seven simulated aerosol properties as stated in Sect. 2.2. There is no vertical dependency here since the method is applied separately in each of the three atmosphere layers as defined in Sect. 2.2. A common requirement for the K-means algorithm is the standardization of the input dataset, due to the fact that input quantities span different orders of magnitudes and can have different units. Since aerosol mass and number concentrations have different units and are characterized by very different numerical values, each of the individual

aerosol properties $x_l$, $l \in \{1, ..., j\}$, are standardized to $xs_l$ assuming the deviation of the data from their respective mean to follow a Gaussian distribution with zero mean and variance of one:

$$xs_l = \frac{x_l - \overline{x_l}}{\sigma_l} (6)$$

where $xs_l$ stands for standardized data, $x_l$ is the original data, $\overline{x_l}$ is the mean and $\sigma_l$ is the standard deviation of this specific aerosol property $l$ calculated from the whole set of samples. The standardization ensures the comparability of the different

aerosol quantities, facilitates evaluating the prominence of individual aerosol properties in the respective regimes, and avoids clustering due to one dominate species but instead focusing on the connection between the different species.



**Figure 1: Simulated climatological aerosol properties for the lower troposphere (surface to ~700hPa) including vertically integrated mass concentration of mineral dust (a), BC (b), sea salt (c), POM (d), SNA (e), vertically integrated particle number concentration of the Aitken mode $N_{akn}$ (f) and of the accumulation mode $N_{acc}$ (g).**





# 3 Results

In this section we present the results of K-means clustering for global aerosol properties in three atmospheric layers as defined in Sect. 2.2. We focus on 4 components: (1) the spatial distribution of the seven individual aerosol properties as inputs for K-means analyses; (2) the evaluation metrics for K-means clustering which support the selection of a proper cluster number $k$; (3) the spatial distribution of classified aerosol regimes; and (4) the characteristics identified for each aerosol regime regarding the data distribution of aerosol properties within each class.

The results of the clustering analyses are visualized in this study using global maps of the cluster distributions. In addition, we show so-called box plots which provide additional statistical descriptions of the data distributions for individual aerosol parameters within each cluster. By comparing the data distributions between individual aerosol parameters and regimes we explicitly analyse the characteristics of each regime.

## 3.1 Lower troposphere clusters

For these clusters, the aerosol mass and number concentrations from the global simulation are vertically integrated from the Earth surface to the model layer which corresponds to about 700 hPa. The resulting spatial distributions are shown in Fig. 1. High mineral dust column masses (up to $1 \times 10^6$ µg/m$^2$) are simulated over the Sahara and in other deserts, while values in other regions are mostly small (Fig.1a). BC column masses are highest in south and east Asia (up to about $3.5 \times 10^3$ µg/m$^2$), due to anthropogenic pollution, and over central Africa (about $2 \times 10^3$ µ g/m$^2$) resulting from intense biomass burning activity (Fig.1b). Peak values of the sea salt column masses over the oceans range between $1 \times 10^4$ µg/m$^2$ and $2 \times 10^4$ µg/m$^2$ (Fig.1c). The pattern of POM columns closely follows that of BC, since the two species share similar emission sources (Fig.1d). Enhanced total masses of sulfate, nitrate, and ammonium (SNA) are noticeable especially for the south of the Eurasian continent (up to $5 \times 10^4$ µg/m$^2$) and the Arabian Peninsula (Fig.1e), which could be due to coal burning for energy production (Klimont et al. 2013) especially in the case of India and China. Column integrated numbers of Aitken mode particles, in the following called Aitken mode number columns, are generally high in the Northern Hemisphere, with large values close to strongly polluted areas (Fig.1f), while biomass burning largely contributes to the accumulation mode number column, which is particularly high in prominent biomass burning regions such as Central Africa and South America (Fig.1g). As expected, aerosol mass and number column show a large spatial variation in the lower troposphere, closely following the geographical distribution of the main emission sources. This variability results in a complex pattern of aerosol regimes as shown below.



**Figure 2: Lower troposphere clustering using K-means. The top panel shows the evaluation metrics SSE (a) and SS (b) vs a *k* range of 2-14. The middle plot (c) highlights the spatial distribution of 10 aerosol regimes for the lower troposphere. The bottom plot (d) shows the data distribution of the 7 considered aerosol properties within the 10 individual aerosol regimes, and cluster names assigned to each cluster based on the analysis of the aerosol data within the respective cluster. The boxplots describe the distribution of data by displaying 5 statistical quantities: the maximum value (top whisker), 75% quantile, median (top of box), median (middle line in box), 25% quantile (bottom of box) and minimum value (bottom whisker) of standardized aerosol parameters that are not outlies. The black dots are outliers which are defined as the data beyond 2.67σ of a normal distribution.**





As explained in Sect. 2.3, K-means classifications are conducted for a range of predefined cluster numbers $k$. The resulting classification is coarse at low $k$, while increasing $k$ leads to increased complexity. At some point, however, the added complexity of the K-means classification does not add further information and therefore a further increase of $k$ is not useful.

Hence, choosing a proper cluster number for the K-means analysis is not straightforward. Here, we use 10 clusters for the lower troposphere based on the K-means evaluation metrics (SSE and SC) and on expert judgement as described above. SSE describes the sum of squared errors from each sample to the respective cluster centre (Eq. 3) and decreases with increasing $k$. For the lower troposphere, SSE decreases rapidly from $k=2$ up to about $k=7$ and then more slowly for larger $k$ (Fig. 2a). The SC is highest at $k=2$, decreases between $k=2$ and $k=4$ and reaches a roughly constant level at $k=5\text{-}11$ (Fig. 2b). The higher the

SC value is, the more similar are the data within the cluster and the more distinct to other clusters. The optimal solution is obtained by minimizing SSE and maximizing the SC. Therefore, taking a balance between small SSE and large SC, we limit the selection of $k$ to 9 to 11. The difference between the 9-cluster and the 10-cluster classification is that one oceanic aerosol regime in the 9-cluster classification is further divided into two clusters in the 10-cluster classification. The 11-cluster classification includes a tiny regime which adds little information with respect to the 10-cluster one. We therefore choose $k=10$

for the classification of aerosol in the lower troposphere.

The resulting 10 aerosol regimes classified by K-means for the lower troposphere are displayed in Figure 2c. These identified major aerosol classes match well the expected regimes in this altitude range. Polar aerosols are classified in cluster 0, while oceanic aerosols are roughly divided between Northern and Southern Hemisphere by clusters 6 and 8, respectively. The large

forests and savannas of Africa and South America are covered by cluster 5 and cluster 1 including major biogenic and fire aerosol sources (e.g., Dentener et al., 2006). Clusters 9 and 3 cover the main desert regions over Sahara and the Arabian Peninsula. Cluster 9 marks the strong dust emission spots, while cluster 3 represents a kind of "background desert" which shows slight influences by aerosol transported from surrounding areas. The regions characterized by strong anthropogenic pollution (Southern and eastern Asia) are assigned to cluster 7, while regions with moderate and low pollution are covered by

cluster 4 and cluster 2, respectively, with the latter often extending to oceanic regions possibly affected by long-range transport of anthropogenic pollution from the continents.

The characterization of the aerosol regimes in the lower troposphere obtained with the K-means method can be further explored and interpreted using the boxplot in Figure 2d. The figure shows the distribution of samples collected within each regime and

several statistical metrics, including maximum, 75% quantile, median, 25% quantile and minimum of the standardized aerosol parameters that are not outliers. We recall the use of a multi-annual mean sample values and the consideration of column integrated values in the lower tropospheric column. The dots are outliers that can be ignored for statistical discussion. They are defined by using +/- 1.5 time of interquartile range of data, which corresponding to data beyond 2.67 sigma of a normal distribution. Note that values on the y-axis are the standardized values (calculated with Eq. 5) but not the absolute value as





shown in Fig.1, in order to do a proper classification with K-means and to compare species with different units and scales. All aerosol properties within cluster 0 (polar regions) have lower values than in the other clusters, meaning that this can be considered as aerosol background, as denoted also in Figure 2d. Low values are found also in clusters 6 and 8, with the exception of sea salt, which has enhanced values: we therefore mark these two clusters as oceanic aerosol. Clusters 6 and 8 are very similar, which explains why they are merged into one cluster if a 9-cluster classification is used. The difference between them are the slightly higher values of aerosol properties other than sea salt concentrations within cluster 6, which points to a more polluted marine regime than in cluster 8, which represents remote oceanic regions. Cluster 1 and 5 cover the major forests and savannas in Africa and South America and downwind areas and are characterized by enhanced POM, BC and $N_{acc}$, which are all typical indicators of strong biomass burning and biogenic activity. The difference between the two clusters is that the enhancement of these quantities is more pronounced in cluster 5 compared to cluster 1. This difference suggests that fresh biomass burning and biogenic aerosol characterize cluster 5, while more aged particles are found in cluster 1 as a result of long-range transport and the subsequent dispersion of the affected air masses in combination with particle wet and dry deposition. Cluster 9 and cluster 3 both have enhanced mineral dust values which agrees with their locations in large deserts or in close proximity to desert regions. Cluster 9 shows much larger mineral dust values and much lower values for the other aerosol properties (in particular SNA and $N_{akn}$) than cluster 3. This suggests that cluster 9 covers the regions of localized strong dust emissions, while cluster 3 includes dust dominated air masses which are mixed with pollution from other regions. The dominance of BC and SNA in cluster 7 matches well with the large pollution characterizing the south and east Asian regions covered by this cluster. Cluster 7 also shows enhanced POM and number concentrations in both aitken and accumulation modes. We therefore name it the enhanced polluted Asian cluster. Clusters 2 and 4 cover large parts of the Eurasian and American continental regions. Cluster 4 is more polluted than cluster 2, but both are relatively clean compared to other continental clusters nearby (e.g., the strongly polluted Asian regions). We refer to these clusters as polluted continental and continental background, respectively. Another important aspect worth noting is that continental aerosol clusters frequently propagate into oceanic regions, showing that this method is also able to capture the long-range transport of pollutants from the emission regions to the relatively clean marine environment. For example, clusters 1, 2, and 3 cover also parts of the middle Atlantic Ocean, cluster 2 also appears over the Pacific Ocean near the west coast of the American continent, and cluster 4 extends over the north western Pacific.

### 3.2 Middle troposphere clusters

The clustering analysis for the middle tropospheric layer uses global aerosol data from about 700 hPa to 300 hPa. As depicted in Fig. 3, this altitude range shows lower values for the column mass and number concentrations (Fig. 1). For example, the column mass of middle troposphere mineral dust ranges from $2 \times 10^3\,\mu g/m^2$ to $3.4 \times 10^4\,\mu g/m^2$ (Fig. 3a) in areas with prominent dust impact, compared to a range of $100\,\mu g/m^2$ to $1 \times 10^5\,\mu g/m^2$ in the lower troposphere (Fig. 1a). This is caused by the decrease of air density during upward transport, by the dilution of the dust load due to mixing with dust-free air masses as well as by possible sinks due to wet deposition. A similar reduction is also evident in the other aerosol properties. The spatial



distribution patterns, however, remain the same between middle troposphere and lower troposphere. However, the overall patterns, in many cases, show a larger spatial extension, caused by long-range transport and dispersion of the respective air masses.


Figure 3: The same as Figure 1 but for the middle troposphere (from ~ 700hPa to ~300hPa).





Due to this dispersion, a less complex clustering is required than in the lower troposphere. In general, we can expect $k$ to decrease with increasing altitude, due to the more uniform spatial aerosol distributions in the upper atmospheric layers. For the middle troposphere, we evaluated K-means classifications with $k$=2 to $k$=8 using the same metrics as applied above (Fig.4 a and b). As for the lower tropospheric case, SSE decreases with increasing $k$, more slowly for $k{\geq}6$. The SC decreases to a minimum for $k$=4 and increases again to a stable level between $k$=6 and $k$=8. The distribution of the major aerosol regimes

becomes very robust at $k$=6, while only minor regimes are introduced at higher values which do not show prominent features. We therefore choose a 6-cluster classification for the middle troposphere.

        In the middle troposphere, the aerosol regimes are more zonally uniform than lower down but the lower troposphere has still a very strong influence on the pattern (Fig. 4c). This is particularly the case for clusters 0, 2 and 5 which appears to be related

to the increasing prevalence of zonal wind patterns in middle troposphere. Clusters 1, 3 and 4, on the other hand, show a stronger influence of the distribution of the emission sources and the transport patterns of the lower troposphere. The statistical analysis of the aerosol properties within each cluster allows to broadly classify the clusters 2 and 5 as middle tropospheric background clusters, and clusters 1, 3, and 4 as middle tropospheric polluted clusters (Fig. 4d). The lowest values of all aerosol properties are found in cluster 5 which can be classified as middle tropospheric background (relatively clean) and covers large

fractions of the southern hemispheric oceans and the polar regions. Cluster 2 is characterized by enhanced sea salt values, while the other aerosol species remain low as in cluster 5. Hence the cluster includes background air enriched with sea salt due to enhanced wind-driven emissions. Cluster 2 mainly covers the intertropical convergence zone (between 20°S and 20°N) with its strong updrafts and the southern hemispheric storm track area around 60°S, which is also an uplift region between the mid-latitude cell and polar cell of the main atmospheric circulation pattern. Due to the strong upward transport in these regions, sea

salt is lifted from the sea surface to the middle troposphere. Cluster 0 is mainly located in the Northern Hemisphere and above the continents: it is characterized by mildly enhanced BC, SNA, POM, $N_{akn,}$ and $N_{acc}$. Similar enhancements of some of these aerosol properties are evident in clusters 1, 3, and 4, but with much larger values. These clusters show similar aerosol characteristics and cover similar regions as their counterparts in the lower troposphere (note however that the algorithm assigns different cluster index numbers between lower and middle troposphere). These three polluted clusters nicely identify three

distinct sources: cluster 1 is mostly affected by the strong emission regions in south and east Asia and southern Europe/Mediterranean, cluster 3 presents a mixture of mineral dust and other pollutions sources, with an evident prominence above large deserts, and cluster 4 is an enhanced carbonaceous/biogenic cluster, with significant coverage over the biomass burning and biogenic sources e.g. in South America and Africa. It occurs also over East Asia with its high anthropogenic emissions of carbonaceous particles. Note that the scaled values in Fig. 2d and Fig. 4d should not be compared directly among

the different atmospheric layers, because the input data for K-means analyses are scaled individually based on the data within each layer.



**Figure 4: The same as Figure 2 but for the Middle troposphere (from ~ 700hPa to ~300hPa).**





**Figure 5: The same as Figure 1 but for the tropopause region (from ~ 300hPa to ~100hPa).**






### 3.3 Tropopause region clusters

The clustering analysis for the tropopause region considers global aerosol data from about 300 hPa to 100 hPa. The degree of spatial dispersion again increases when compared to the lower layers. Therefore, the distributions become more homogeneous

than in the middle and lower troposphere (Fig.5). The maximum values of the five aerosol mass columns (mineral dust, BC, sea salt, POM, SNA) are lower in the tropopause region (Fig. 5) than their background value in the lower troposphere (Fig.1). For example, the maximum mineral dust mass column in the tropopause region amounts to about $1 \times 10^3 \, \mu g/m^2$, this value is about the minimum value of mineral dust in the lower troposphere. Although aerosol mass columns in the tropopause region are generally small and a high degree of dispersion is reached, the spatial patterns for mineral dust, BC, POM and SNA are

still closely related to those in the lower troposphere. This demonstrates that local upward transport of aerosols from the Earth's surface to the tropopause region is efficient in areas showing enhanced dust concentrations. However, this does not fully apply to sea salt, which reaches high values only in the tropics corresponding to regions of strong convection over the oceans in the tropopause region (Fig. 5c). With regard to the aerosol number columns, the effects of vertical and zonal transport appear to be more complex. While the accumulation mode particle number shows a similar behaviour as the mass loadings, the Aitken

mode particle number column appears to be strongly influenced by new particle formation in the tropopause region. Hotspots of the particle number occur particularly over regions of enhanced gaseous pollution which provides aerosol precursor gases, such as $SO_2$, leading to aerosol nucleation and growth favoured by the clean environment of the tropopause region.

As mentioned above and confirmed by the homogeneous characteristics of aerosol in the tropopause region shown in Fig. 5, a

more simplified clustering can be applied in this layer, reducing $k$ to less than 6. The SSE of K-means clustering for the tropopause region (Fig. 6a) shows a similar structure as in the middle troposphere (Fig. 4a), with noticeable convergence from about $k=6$. The SC reaches a maximum for $k=4$ and $k=5$ (Fig. 6b). The combination of these two metrics suggests $k=5$ as the proper choice for the K-means classification for the tropopause region. The resulting 5 clusters are shown in Figure 6c. Large parts of the tropopause region belong to cluster 1, which covers the whole Polar regions and most of the Southern Extra-tropics.

The second largest cluster is cluster 2, which covers a large part of the northern extra-tropics and about half of the tropical ocean regions, with the other half mostly covered by cluster 3. Cluster 0 and 4 cover a small portion of the continents including central Africa, the Saharan region as well as tropical and subtropical Asia. Figure 6d highlights the aerosol characteristics for each cluster of the tropopause region. Cluster 1 shows the lowest values for all aerosol properties which suggests to characterize it as tropopause region background. Note that in the pole regions, the pressure levels considered here are mostly located in the

stratosphere, and therefore contain comparably clean air. Cluster 3 has similarly low values for all species except for sea salt, which is significantly enhanced due to upward transport in the intertropical convergence zone. Hence, we denote it as the tropopause region enhanced sea salt cluster. The slightly enhanced $N_{acc}$ in cluster 3 relative to the cluster 1 is probably caused by new particle formation. Cluster 2 shows slight increases for all aerosol properties relative to cluster 1, but being still lower





than in the other clusters. We therefore define cluster 2 as the tropopause region mildly polluted cluster. Cluster 0 features

strongly increased mineral dust accompanied by slight increases in BC and SNA. Therefore, it can be termed tropopause region dust/polluted cluster. This is also supported by its geographical location over the Sahara and the Middle East where mixtures of desert dust with anthropogenic pollution could be expected. Cluster 4 shows strongly enhanced BC, SNA and POM, and mildly enhanced mineral dust which suggests to term this regime tropopause region polluted/mixed cluster. On the one hand, it is strongly influenced by the biomass burning and biogenic aerosol sources over central Africa and South America. On the

other hand, it shows also relevant coverage over East Asia, resulting from the strong pollution sources in these regions. Note that there are many similarities between the aerosol regimes of the tropopause region and the mid troposphere (Fig. 4), especially for clusters 3 and 4, which are largely controlled by efficient updrafts. Hence these clusters correspond also well to lower tropospheric aerosol regimes of similar characteristics occurring in the same regions (Fig. 2).



**Figure 6: The same as Figure 2 but for the tropopause region (from ~ 300hPa to ~100hPa).**




## 4 Discussion

This study demonstrates the successful application of the K-means algorithm for the classification of global aerosol
climatological regimes in model simulation output. It provides quantitative information about the aerosol regimes across the
globe and at three altitude ranges, from the surface to the tropopause region. The clustering analysis performed by the algorithm
allows to systematically characterize many aerosol properties (such as spatial distribution, composition and extension) in a
single index, thus facilitating the analysis of the output of global model simulations. This study represents a first attempt to
apply the clustering method to global aerosol modelling. However, it has of course limitations and potential for improvements.
These are discussed in the following, together with suggestions for possible applications of the presented method.

The K-means method has advantages and disadvantages in performing classification tasks. The advantage is that it does not
require prior classification knowledge or training data (Hastie et al., 2009). In cases where no detailed concepts for a pre-
definition of aerosol classes based on primary aerosol model parameters can be provided, using K-means is a proper approach.
The disadvantage is that the K-means method is sensitive to data variability. Our calculations demonstrated, for instance, that
a too high variability resulting from the consideration of temporal variation complicates the K-means clustering. Beyond the
analysis of multi-annual means, we attempted to classify global climatological seasonal data which include the variability in
the time dimension from four seasons. This attempt resulted in complications in the classification across the four seasons, since
the seasonal variations, in many cases, are larger than the differences between the specific clusters, which leads to large changes
in the characteristics of the clusters and their spatial extent from season to season. This shows that the K-means method
discussed here does not work well for analysing the data variability across time and space simultaneously, as the interpretation
of the resulting classification would be challenging. To overcome this limitation, we removed the variability in the time
dimension in this study by considering multi-year averages of the model output, thereby setting a focus on classifying the
spatial distribution of long-term climatological aerosol regimes. Possible inter-annual and seasonal variability of aerosol
properties could alternatively be discussed on the basis of the climatological regimes analysing the internal changes of aerosol
properties within the climatological clusters obtained by K-Means.

Another limitation of this study is inherent in the data to which the algorithm is applied. Our clustering analysis is based on
data from a model simulation performed with EMAC-MADE3. Despite the detailed evaluation of this model (Kaiser et al.
2014, 2019) and in particular of the simulation considered here (Beer et al., 2020), some model biases and deficiencies remain
and could affect the outcome of the clustering algorithm. That is, the clusters derived from the model output could deviate
from their appearance in the real atmosphere. Applying the same algorithm to observational data, on the other hand, is not
feasible, since no dataset including all relevant chemical and microphysical aerosol properties with global coverage and vertical
resolution exists. Vertically resolved data are available from in-situ aircraft-based measurements, but their geographical
coverage is limited and they are often not representative on a climatological scale. Satellite data could in principle provide





global coverage, but they usually comprise optical aerosol properties, such as aerosol optical depth or aerosol extinction (e.g., Popp et al., 2016). Optical aerosol quantities could be used for classification (e.g. Groß et al, 2015) but the resulting classes do not necessarily reflect the details of aerosol composition and size. In this context, using global model simulation data for classifying global aerosol regimes is an appropriate strategy, also considering the fact that the distribution of data is more

important than their actual value for K-means clustering. Model systematic biases are not necessarily related to wrong data distribution. For example, systematic model biases in model parameterizations (e.g. overestimation/underestimation) cause errors in the absolute values of simulation variables, but these errors are cancelled out when the data are normalized for the K-means analysis. Studies have shown that models generally capture the spatial patterns of aerosol properties quite well but their actual values are biased (Mann et al. 2014; Koffi et al. 2015; Kaiser et al. 2019; Beer et al. 2020). However, simulation biases

in the spatial patterns will change the identified regimes. The extent of this change needs to be further investigated in future studies.

Despite its limitations the K-means method presented in this study is a very helpful tool to analyse and interpret the huge amount of aerosol data generated by global simulations including detailed descriptions of the size-resolved aerosol

composition. The method has a wide application potential. Since the algorithm identifies aerosol regimes by minimizing the variance within each cluster, the aerosol properties within a cluster are similar to each other. This implies that aerosols can be treated cluster-wise instead of grid-point-wise, thus reducing the amount of data required to describe the global aerosol population. The possible applications of this method include (but are not limited to) the following:

1.  Investigating and correcting model systematic biases using observational data is an important aspect in aerosol model
development. However, it is often challenging due to the limited temporal and spatial coverage of observational data. Using the K-means algorithm to identify major aerosol regimes allows to simplify bias-adjustment approaches, since even spatially limited observations within a given cluster can be used to adjust the biases in other regions of that regime. We would address only systematic model biases which occur nearly homogenously throughout the whole cluster, but not purely local model discrepancies. The bias-adjustment for global aerosols remains nevertheless
difficult, since it requires a systematic compilation and homogenization of observational aerosol data from different sources, instruments and regions, and requires the consideration of various observational uncertainties. This is planned for a follow-up study.

2.  The identified aerosol clusters can be used as first order criteria for satellite retrievals. Some satellite retrieval algorithms (Holzer-Popp et al. 2018; Kahn and Gaitley, 2015) first calculate aerosol optical depth for several pre-
defined aerosol types/compositions with top of atmosphere reflectance look-up tables, and then select from the different aerosol types in the atmosphere the best spectral or multi-angular fit between calculated and observed microphysical and optical top of atmosphere reflectance. This is a time-consuming process since a large number of different aerosol types and composition needs to be tested (e.g., 36 or 74 mixtures) without any a-priori pre-selection.



By applying the results of the clustering method presented here the characteristics of each aerosol regime could be used to dismiss unrealistic guesses before applying the retrieval algorithm, thus reducing the computing time.

3. Our results could provide data for training other supervised machine learning algorithms. K-means is chosen in this study because a priori definition of aerosol classes is not straightforward since it would require a thorough analysis of the prevailing aerosol regimes in the model output. This however is intended to be achieved with K-means. But if the prevailing aerosol regimes are known from the K-means results, it is possible to prepare training datasets for other supervised machine learning algorithms for further, more detailed classifications, e.g. using random forest or neural network approaches.

4. The planning of future observational campaigns could benefit from model-based cluster analyses, as they provide useful information on aerosol characteristics in different regimes. Based on this information, campaign planners could easily identify regions of interest regarding specific aerosol properties or types, for example focusing on aerosol from specific sources (e.g. mineral dust from deserts or particles from biomass burning regions).

5. Possible long-term aerosol trends could be analysed by comparing the distribution of clusters calculated for different periods (e.g. pre-industrial, present-day conditions and future scenarios), also providing insights for the validation of climate and air quality measures.

## 5 Summary and outlook

In this study, we apply the K-means algorithm to classify climatological aerosol regimes across the atmosphere, based on seven primary aerosol properties simulated with the EMAC-MADE3 global aerosol model. These properties include mass concentration of black carbon, mineral dust, sea salt, particulate organic matter, the sulphate/nitrate/ammonium system, and the aerosol number concentrations of the Aitken and accumulation modes. K-means classifies the model data by means of a cluster analysis based on a minimization of the variances, so that data within a respective cluster are similar to each other but different to that in other clusters. K-means has been proven to be a powerful classification tool and is especially useful when prior classification knowledge is not available. We apply K-means to quantitatively identify global aerosol regimes and explain the characteristics of the classified regimes regarding their location, extent, and specific aerosol properties. This study represents the first application of this algorithm for the classification of the global aerosol. The results show that in the lower troposphere, the aerosol regimes are largely controlled by emissions. Different aerosol clusters are identified, characterized by biomass burning or biogenic activity, mineral dust, anthropogenic pollution, background conditions, as well as a mixture of these different types. Several continental clusters propagate over the oceans due to long range transport of the affected air masses. The algorithm classifies the oceanic regions in two major clusters, with a moderately polluted northern hemisphere and a cleaner southern hemisphere. In the mid troposphere and the tropopause region the aerosol regimes are more zonally uniform than lower down, but the lower troposphere has still a very strong influence on the pattern. Evidences are three polluted

clusters occurring over Africa, southern and eastern Asia. Due to efficient vertical dispersion these clusters are present at all altitude levels and show similar characteristics from the surface to the tropopause region. The classification of aerosol has a wide spectrum of potential applications. We have suggested several possible future applications that could benefit from this classification scheme. These include identifying model biases and conducting bias-adjustment, preparing training data for

other supervised classification algorithms, simplifying satellite retrieval processes, and supporting campaign planning.

**Acknowledgements**

We are grateful to Dr. Ulrike Brukhardt (DLR, Germany) for her suggestions on an earlier version of this manuscript. We

thank Dr. Thomas Popp (DLR, Germany) for his great help on discussing the potential usage of aerosol regimes for satellite retrievals. We are thankful to the developer of Python package Scikit-learn for providing this excellent machine learning package. The model simulations and data analysis for this work used the resources of the Deutsches Klimarechenzentrum (DKRZ) granted by its Scientific Steering Committee (WLA) under project ID bd0080.

**Financial Support**

This work was supported by the German Federal Ministry for Economic Affairs and Energy - BMWi (project "Digitally optimized Engineering for Services" – DoEfS; contract no. 20X1701B), the DLR space research program (project "Innovative methods for analyzing and evaluating changes in the atmosphere and the climate system" - MABAK), and the DLR transport program (project "Transport and Climate" – TraK).


**Code and data availability**

Documentation of python package Scikit-learn is available at https://scikit-learn.org/stable/.

The model simulation data analyzed in this study are available at https://doi.org/10.5281/zenodo.3941462 (Beer, 2020). The cluster analysis code used in this study is available at https://doi.org/10.5281/zenodo.5121180


**Author contributions**

JL conceived the study, implemented the clustering methods and wrote the paper. JH, MR and CB contributed to conceiving the study, the interpretation of the results and the text. CB performed the simulation used in this study.





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
