# Peer review of "An aerosol classification scheme for global simulations using the K-means machine learning method"

_Geoscientific Model Development, 2021_

## Author Comment (AC2)

**Replies to Referee's Commends**

Reviewer comments are highlighted as italic text. Our responses are shown as plain text. Texts the modified in manuscript are highlighted in red color.

**Response to Reviewer #1**

\*\*\*\*\*\*\*\*\*\*\*\*\*\*\*\*\*\*\*\*\*\*\*\*\*\*\*\*\*\*\*\*\*\*\*\*\*\*\*\*\*\*\*\*\*\*\*

*Review of Li et al.*

*Li et al. present research exploring the application of the K-means clustering method to simulated aerosol data. The authors claim that this method allows for the identification of aerosol regimes and demonstrate the relationship between these regimes and known aerosol sources and property distributions. While the application of clustering methods like K-means show great potential in exploring atmospheric composition, I cannot recommend this paper for publication at this time due to major methodological flaws and a lack of novel scientifically valid results. Major issues are summarized below.*

Response: We thank the reviewer for his/her remarks on our study. Please find our replies to the issues raised below.

*Major Issues*

*Data Standardization*

*Gaussian standardization as described on line 210 is inappropriate for much of the data used here. Aerosol number and mass concentrations typically vary logarithmically (e.g. Figure 1a), and standardization should reflect this variability. Improper standardization can lead to spurious clustering and limits the interpretability of the results.*

Response: The referee raises the correct point that the data analyzed in our study do not follow a Gaussian distribution in many cases. Our statements at lines 210-211 ("…assuming the deviation of the data from their respective mean to follow a Gaussian distribution…") was indeed misleading and we have corrected it in the revised manuscript to "…by subtracting their respective mean and dividing each value by its respective standard deviation.".

We would like to point out, however, that the purpose of this standardization is to weight the different input quantities equally with respect to each other, i.e., to harmonize the concentration values of the different quantities to values of similar magnitude ranges, before applying the K-means algorithm. It is not necessary to assume that the data follows a normal distribution to apply such standardization. Our standardization procedure can be applied to any data, since it does not change the underlying distribution of the data.

We agree with the reviewer's statement that "Improper standardization can lead to spurious clustering and limits the interpretability of the results." The choice of the variance applied for standardization, for instance, could potentially have an effect on the clustering (e.g, due to skewed distributions or simply due to the relatively large number of sample points potentially causing complex multimodal distributions). In order to investigate this in more detail, we have extended the paper by a dedicated section (Sect. 4.1), where the impact of alternative scaling

methods on our results is analyzed, comparing their advantages and disadvantages in the context of the specific data targeted in our work. From this additional analysis, we conclude that "(1) The standardization which we use for this study (S1) simply scales the values of aerosol properties but it does not change the underling distribution of the raw data; (2) the most important criterion for K-means data preprocessing is that the data of different properties should be scaled to a comparable range so that they are more or less equally weighted; (3) S1 is the best choice; and (4) the 'outliers' in the data distribution are important for aerosol clustering." Please see the detailed descriptions of these conclusions in Sect. 4.1.

After this analysis, we are more confident that our approach is robust and that it is not affected by methodological flaws.

*K Value Selection*

*The authors well describe two metrics for selecting the value of K, or number of clusters in the manuscript. However, the quantitative metrics are ignored in favor of "expert judgement" (e.g. line 266) when selecting the number of clusters in section 3.1 and 3.2. In both sections, the SSE plots (Figures 2a and 4a) do now show a distinct elbow and the maximal silhouette coefficient is at 2 clusters. Both of these figures indicate that in the standardized dataset used here there are no strong natural clusters and the applicability of the K-means algorithm should be revisited entirely.*

Reponses: In our original manuscript, the choice of the number of clusters is already well described in detail. It is motivated in Sect. 2.3 and well justified in Sect. 3. The combination of both metrics is the major base of our selection. "Expert judgement" is used only for further refinement and/or plausibility checks.

Selecting the number of clusters is one of the most difficult tasks in cluster analysis and there is no ideal solution for that. There is not always a distinct elbow in SSE and the maximal silhouette coefficient does not always indicate a best value for k. As an example, in the lower tropospheric case of our study, the maximum of the silhouette coefficient occurs at k=2. However, the corresponding SSE value is large which excludes a selection of k=2. Therefore, we use a combination of these two metrics to support our decision. See also our discuses on the plausibility check of the resulting clustering in the Sect. 3.

To address this point, we added the following sentence in the manuscript "Selecting the number of clusters k is one of the most challenging tasks in cluster analysis. Researchers developed many different approaches to select k but there is no standard solution which can be generally applied (e.g. Rousseeuw 1987; Sugar and James 2011; Amorim and Hennig, 2015)." The "expert judgement" used in the original manuscript is replaced by "a plausibility check of the obtained clusters", which we believe is more appropriate.

Furthermore, we included a new section (Sect. 4.2) to compare K-means with an alternative unsupervised clustering algorithm HAC, as also suggested by the Reviewer #2. Although HAC uses a different strategy to choose k, the selection of k from HAC agrees well with our selection for K-means, which further supports the choices of k in the K-means applications of this study. Please see the new Sect. 4.2 in the revised manuscript for details.

*Scientific Results*

*After applying the k-means clustering (with the major limitations outlined above) the authors do not find novel or (in some cases) scientifically valid results. Despite the claim that "specific aerosol characteristics for the predominate regimes are not known a priori" (line 91), a large body of research exists quantifying the regimes and characteristics of aerosol in various regions. The results of this work are at best consistent with that prior knowledge (e.g. the importance of emissions for controlling aerosol regimes in the lower atmosphere). In other cases, the results strongly disagree with prior knowledge in ways that are not adequately addressed in the text.*

*For example, in Figure 2 the clustering analysis shows that nearly all of western Europe is in a "background continental" regime, despite several major anthropogenic aerosol sources (i.e. Paris, London, Benelux). In the same clustering analysis, largely remote areas of Asia and the Middle East are classified in the same aerosol regime as Los Angeles, California, despite the very large differences in aerosol characteristics that are known over these areas. Additionally, the major dust source of the Gobi Desert is not present in any of the dust clusters. The large differences in the results from this manuscript and the existing literature on aerosol regimes potentially indicate larger issues in the results of the clustering algorithm application and are not addressed in sufficient detail in the manuscript.*

Response: We do not agree with these comments due to the following three reasons.

(1) The referee points out that highly polluted hotspots like Paris or the Benelux area are not visible in the clusters. However, since we use a global model with about 200 km horizontal resolution, it is not possible to resolve such small (local scale) features. EMAC-MADE3, as every global aerosol model, represents large scale means which are the basis for quantifications of aerosol-climate effects rather than for addressing, e.g., local air pollution issues. In addition, analyzing long-term means does not highlight specific pollution episodes. Also note that pollution in Europe is not as prominent as, e.g., in some parts of Asia. Hence, from a global modelling perspective, one cannot expect to see such hotspots in the model data. We would also like to stress that the model has been extensively evaluated in several previous studies (e.g. Aquila et al. 2011; Righi et al. 2013; Kaiser et al. 2019; Beer et al. 2020), including comparisons with station measurements in Europe, and was found to be in good agreement with the observations. The Gobi Desert is not a very pronounced dust emission region in our simulation (Figure 1a of the manuscript). Comparisons with other dust emission assessments (e.g. Dentener et al., 2006) reveal that the dust emission strength of the Gobi Desert as represented in the dust emission parameterization applied here is at the low end of comparable global quantifications. Hence, it is not surprising that the Gobi Desert has only a weak influence on the clustering. In contrast, the Sahara, the largest dust source on Earth, is well captured by our analysis. In summary, one cannot expect the K-means algorithm to capture features, which the model itself cannot represent due to its global nature or which are not captured by the underlying emission parameterizations. On the basis of the given model output data, the clustering algorithm generates reasonable results.

We added the above arguments to Sect. 4.3 of the manuscript, where we discuss the strength and limitation of global simulations: "The major goal of this study is the development of a clustering method to complement classical approaches for analyzing and interpreting global aerosols model output. In order to put the demonstration results of the method presented in Sect. 3 in the right context, strengths and limitations of global aerosol simulations are discussed in the following.

Extensive evaluations have been conducted in previous studies to investigate the potential of global aerosol simulations and their limitations (e.g. Textor et al 2006; Lauer et al, 2007; Bauer et al 2008; Koch et al 2009; Mann et al., 2010, 2014; Pringle et al., 2010; Aquila et al 2011; Huneeus et al 2011; Kirkewåg et al. 2013, 2018; He and Zhang, 2014; Koffi et al. 2015; Lee et al 2015; Michou et al 2015; Kaiser et al. 2019). A major deficiency of global aerosol simulations is their inability to resolve small scale and localized processes, largely as a result of the computational challenges and the chemical complexity allowing for only coarse grid resolution in global models. Our clustering analysis is based on data from a global model simulation performed with EMAC-MADE3. The data used has a spatial resolution of about $1.9° \times 1.9°$ in latitude and longitude and can therefore not reproduce smaller-scale features, as for instance aerosol pollution on the scale of specific cities. However, the focus of the present study is the analysis of large-scale global climatological aspects with high relevance for simulating aerosol climate effects. Investigating localized aerosol phenomena and their temporal evolution, which would be of particular relevance for air pollution aspects, is not the intention."

Additionally, we emphasize that global simulations mostly capture the aerosol spatial distribution well by including the following sentence in the manuscript: "Global aerosol simulations mostly capture the major large-scale spatial patterns of aerosol properties well. For the EMAC-MADE3 model applied here this was demonstrated by Kaiser et al. (2019) and Beer et al (2020). Hence also the clustering results can be expected to show the major large-scale features of the global aerosol distribution."

We conclude at the end of Sect. 4.3: "The extensive evaluation performed in the existing global aerosol model studies, considering very large numbers of aerosol-related quantities represented in the simulations, is often difficult to interpret. This, in turn, suggests that new analysis methods, for instance, treating aerosols as groups as presented in this study, are in demand. Although aerosol classification is developed in this study primarily for evaluation purposes, the results of aerosol classification from the global model output potentially provides valuable insights for aerosol research, taking the advantages and limitations of global aerosol simulation into consideration."

Please see also the discussions regarding other limitations and strengths of global aerosol simulations integrated in the Sect. 4.3 of the revised manuscript. We have updated the abstract and conclusions accordingly.

We agree that choosing the name "background continental" for the aerosol cluster covering Central Europe and other regions containing also significant pollution sources is misleading. We therefore renamed it to "weakly polluted continental". For consistency, we also renamed the cluster "polluted continental" to "moderately polluted continental".

(2) We would also like to emphasize that GMD is a journal focusing on methodological aspects and that our manuscript indeed has the major focus on the application of a method, i.e. the clustering algorithms (K-means and now also HAC for comparison) applied to global aerosol model output, showing the advantages and disadvantages of this approach and discussing possible limitations and applications. The primary focus is not to answer specific atmospheric science questions as, for instance, on anthropogenic air pollution issues or the effects of natural particle emission hotspots as the Gobi Desert. Here we simply present a new method as a basis for application in future studies dedicated to specific aspects of the global aerosol distribution and related aerosol-climate effects. Despite this primary methodological focus, the referee misses novel scientific results and therefore questions the relevance of the study. Beyond the points discussed above, we would like to stress that the clustering results highlighted in the manuscript for demonstration of the method's feasibility already provide some new insights on the global aerosol distribution (see point 3 for more details). Arguments on why the new method is valuable and worth publishing are discussed very thoroughly in the introduction section. In particular, the growing complexity of global aerosol models leads to an increasing number of parameters describing, e.g., the aerosol number concentration, size distribution and composition. This complexity makes the analysis, evaluation and interpretation of the model results a challenge. Hence, we present a relatively simple yet powerful tool to facilitate the analysis of such global aerosol model results, i.e. grouping them into different clusters, each characterized by specific properties. This method can be beneficial for many future studies and also for other modelling groups.

(3) With regard to the comment on the "lack of novel scientifically valid results" of our manuscript, we would like to stress the following. The value of our scientific results is stated already in the abstract and further summarized in the conclusion section. As exemplary results we present the distribution and extent of aerosol regimes from the surface to the tropopause region for three different altitude levels, and discuss possible uncertainties also in view of the intrinsic limitations of global modelling (as mentioned above). To our knowledge, such kind of analysis has not been conducted before. We agree with the referee that there is much information available on aerosol properties in specific regions. However, there is much less information available on the extension of the corresponding aerosol regimes from a global, climatological perspective, in particular with regard to the mid troposphere and the tropopause region. In this context, the clustering approach introduced here already provided valuable new insights and has a large potential to further deepen these in future studies.

**Response to Review #2**

\*\*\*\*\*\*\*\*\*\*\*\*\*\*\*\*\*\*\*\*\*\*\*\*\*\*\*\*\*\*\*\*\*\*\*\*\*\*\*\*\*\*\*\*\*\*\*\*\*\*

*Dear authors*

*Thank you for submitting this work for review. Firstly, I would agree that it is interesting to evaluate methods for grouping model outputs into distinct clusters, if only for the purposes of an additional diagnostic. As we know, aerosol formulations within large scale models are not*

*ideal; largely as a result of the computational challanges and chemical complexity retained within earth system modules. We should not shy away from presenting results that are negative in the sense of clearly demonstrating limitations of proposed methdologies, however simple or complex they are. All this aside, I do feel this paper requires more work and clarity before it could be published in GMD as per the following discussion.*

Response: We appreciate the very helpful comments and constructive suggestions from the reviewer, and we thank the reviewer for her/his support. We have implemented all suggestions, which strongly improved the quality of the manuscript.

*Whilst K-means is a fast technique, it does rely on a number of assumptions on the data in focus. This includes the presence of clusters with equal size and density. Limitations also include sensitivity to outliers and poor accuracy scaling with increasing dimensions. There are studies that modify k-means to increase its performance, including the use of neural network techniques that act as a non-linear dimension reduction approach to generate K-means 'friendly' spaces. Have you used PCA to assess any changes in cluster properties? Presently there is not enough information in the manuscript on the distribution of each metric used in the clustering. Plotting the distribution of values for each metric would help guide the reader to better consider the appropriate choice of pre-processing. I understand that choice of clustering technique can be at the whim of the investigator, but there should be a level of data exploration that helps form the narrative around the relevance of the results.*

Response: We have addressed this shortage of data explorations by extending the manuscript with several new discussions on methodological aspects in Sect.4.

Regarding the comment on "Limitations also include sensitivity to outliers and poor accuracy scaling with increasing dimensions", we have compared our reference scaling method with 4 alternatives and have investigated how these different scaling procedures affect K-means clustering (new Section 4.1). The fourth conclusion drawn from this analysis addresses the above comment, in particular: "(4) The 'outliers' in the data distribution are important for aerosol clustering. We tested this by applying the base-10 logarithm to the original (skewed) distribution, resulting in a more gaussian-like distribution (Fig. 7, third column), thus removing the outliers. When applying the K-means algorithm with this method, several polluted clusters vanish (compare Fig.8 a and b). Although the basic structure of clusters is still visible, some important information is not captured with the S2 method. For the purpose of the present work, these high values in the data distribution should not be interpreted as outliers in the general sense, i.e. indicating noise and wrong information, which could hinder K-means clustering, but are rather due to the intrinsically large spatial differences of aerosol properties across the globe and they do provide useful information on the data set. It is also important to recall, that we consider climatological data averaged over a long-term period (14 years), which already excludes unrepresentative high values in the aerosol distribution."

Regarding the comments on using "neural network techniques that act as a non-linear dimension reduction approach to generate K-means 'friendly' spaces" and the use of "PCA to assess any changes in cluster properties", we are not able to implement such extensive techniques in the short term. We would need to investigate the relationship between PCA and K-means more thoroughly. This could be the subject of future studies. On the other hand, as we stated in the previous paragraph, K-means' sensitivity to outliers seems not to be a problem for this study.

Section 4.1 provides additional conclusions drawn based on our data explorations, please see the whole Section 4.1 in the revised manuscript for details.

*With this in mind, I would like to see more discussion on the benefits and limitations of general unsupervised techniques earlier on in the manuscript. You do note these at the end of the paper, but I feel these should be discussed earlier.in the summary and outlook, you state that 'K-means has been proven to be a powerful classification tool' - is this with regards to this study or generally? Please clarify statements like this. Given the known limitations, it is difficult to agree with this.*

Response: We agree that the statement "K-means has been proven to be a powerful tool" is not appropriate and we have deleted it from the manuscript. We now state that "K-means is especially useful when prior classification knowledge is not available."

A sentence stating the advantage of unsupervised classification schemes was already included in the introduction of the original manuscript: "K-means belongs to the class of unsupervised machine learning algorithms. This is especially useful when the classification criteria are unknown, as in the case of aerosol classification where the specific aerosol characteristics for the predominant regimes are not known a priori."

We have extended it by the following sentence to address the difference between supervised and unsupervised classifications and the limitations of K-means: "In comparison with supervised classification algorithms which requires substantial prior knowledge of classes, an unsupervised classification is relatively easy to use, but it requires the identification and labelling of the resulting clusters after the classification. The common known limitations of K-means include the presence of clusters with equal variances and its sensitivities to outliers."

*If K-means is used primarily due to its computational performance, please state this with a discussion on the data challenges you have. It seems you do not have a significant amount of gridded data [~18K points?] to cause a problem with regards to computational cost. However, you mention 'the huge amount' of data from global simulations toward the latter stages of the manuscript. How many points did you begin with?*

Response: We used 18432 data points in this study (96 latitude $\times$ 192 longitude points), each containing information about 7 different aerosol properties. "The huge amount" of data is stated in the manuscript to describe the volume of the global aerosol simulation output in general, which includes fine temporal resolution (e.g. hourly) of 3D fields and a huge number of variables. In the future, we would like to apply our classification to more complex cases.

Regarding the computational performance, we included the following statement in section 4.2 of the revised manuscript (comparison of HAC and K-means): "Another aspect to be considered when comparing these two clustering algorithms are the computational expenses. K-means is a fast algorithm, its computing cost does not scale considerably with sample size or dimensions. HAC has a higher demand on computing time than K-means, especially when the sample size is large. For a sample of size $n$, the computing cost of HAC scales approximately as $n^2$ (Dasgupta, 2016; Roy and Chakrabarti, 2017). This is because the hierarchical clustering considers all possible merges at each step, resulting in a rapidly increasing computing time for larger samples. However, HAC has a hierarchy structure (dendrogram) which is more informative and straightforward for deciding on the number of clusters. For this study, both methods provide similar results. Considering further applications

of clustering in more complex situations, we chose K-means primarily due to its computational performance."

*Following on from this, I would suggest you provide comparison with another method for clustering before reaching a set of conclusions as to the viability of K means. This could be hierarchical agglomerative clustering [HCA], with appropriate pre-processing as per the approach used with K means. If your dataset is indeed ~18K points this will not take long to compute. If this is a concern you may find significant improvements from the fastcluster package which can be called using the same syntax as those within Scipy: http://danifold.net/fastcluster.html. Just to re-iterate here: If you wish to demonstrate the performance of K-means then the justification and limitations, according to the volume and properties of your data, must be clear. Comparison with HCA, for example, may give this study a useful balance.*

Response: We are very grateful for this suggestion. We have provided a comparison of K-means with HAC in the new section 4.2. The cluster distribution of K-means and HAC clustering show a good agreement with only small differences. Hence, the results of the classification appear to be robust. For details, please see the corresponding discussion in section 4.2 of the revised manuscript.

*Please also add the balance of limitations in the abstract. The statement 'A markedly wide application potential of the classification procedure is identified and further aerosol studies are proposed which could benefit from this classification.' requires that additional body of work.*

Response: We have deleted this sentence in the abstract "A markedly wide application potential of the classification procedure is identified and further aerosol studies are proposed which could benefit from this classification." and replaced it with "The results of this analysis need to be interpreted taking the limitations and strengths of global aerosol models into consideration. On the one hand, global aerosol simulations cannot estimate small-scale and localized processes due to the coarse resolution. On the other hand, they capture the spatial pattern of aerosol properties on the global scale, implying that the clustering results could provide useful insights for aerosol research. To estimate the uncertainties inherent in the applied clustering method, two sensitivity tests have been conducted i) to investigate how various data scaling procedures could affect the K-means classification and ii) to compare K-means with another unsupervised classification algorithm (HAC, i.e. Hierarchical Agglomerative Clustering). The results show that the standardization based on sample mean and standard deviation is the most appropriate standardization method for this study, as it keeps the underling distribution of the raw dataset and retains the information of outliers. The two clustering algorithms provide similar classification results, supporting the robustness of our conclusions. The classification procedures presented in this study have a markedly wide application potential for future model-based aerosol studies."

*Please also provide the file used to perform the clustering. I have reviewed the zenodo instances for both model output and cluster script, but there seems to be a significant disconnect between the global model output store and a file used in the cluster procedure. Please provide at least so information on how one can extract the relevant files to ensure reproducability.*

Response: Thank you for pointing this out. We have provided the data files used to perform the clustering, and created a new doi at: https://zenodo.org/record/5582338

The corresponding text in the Code and data availability section has been modified to "The information on the simulation setup can be found on the zenodo repository for the Beer et al. 2020 paper (https://doi.org/10.5281/zenodo.3941462). The data and script used in this study is available at https://zenodo.org/record/5582338."

**Citation**: https://doi.org/10.5194/gmd-2021-191-RC2

**References:**

Amorim, R. C. D. and Hennig, C: Recovering the number of clusters in data sets with noise features using feature rescaling factors, Inf. Sci., 324, 126-145, doi: 10.1016/j.ins.2015.06.039, 2015.

Aquila, V., Hendricks, J., Lauer, A., Riemer, N., Vogel, H., Baumgardner, D., Minikin, A., Petzold, A., Schwarz, J. P., Spackman, J. R., Weinzierl, B., Righi, M., and Dall'Amico, M.: MADE-in: a new aerosol microphysics submodel for global simulation of insoluble particles and their mixing state, Geosci. Model Dev., 4, 325-355, doi:10.5194/gmd-4-325-2011, 2011.

Bauer, S. E., Wright, D. L., Koch, D., Lewis, E. R., McGraw, R., Chang, L.-S., Schwartz, S. E., and Ruedy, R.: MATRIX (Multiconfiguration Aerosol TRacker of mIXing state): an aerosol microphysical module for global atmospheric models, Atmos. Chem. Phys., 8, 6003–6035, doi:10.5194/acp-8-6003-2008, 2008.

Beer, C. G., Hendricks, J., Righi, M., Heinold, B., Tegen, I., Groß, S., Sauer, D., Walser, A. and Weinzierl, B.: Modelling mineral dust emissions and atmospheric dispersion with MADE3 in EMAC v2.54, Geosci. Model Dev., 13, 4287-4303, doi:10.5194/gmd-13-4287-2020, 2020.

Dasgupta, S.: A cost function of similarity-based hierarchical clustering, The 48th Annual ACM SIGACT Symposium, doi: 10.1145/2897518.2897527, 2016.

Dentener, F. and 16 co-authors: Emissions of primary aerosol and precursor gases in the years 2000 and 1750 prescribed datasets for AeroCom, Atmos. Chem. Phys., 6, 4321-4344, doi:10.5194/acp-6-4321-2006, 2006.

He, J. and Zhang, Y.: Improvement and further development in CESM/CAM5: gas-phase chemistry and inorganic aerosol treatments, Atmos. Chem. Phys., 14, 9171–9200, doi:10.5194/acp-14-9171-2014, 2014.

Huneeus, N., Schulz, M., Balkanski, Y., Griesfeller, J., Prospero, J., Kinne, S., Bauer, S., Boucher, O., Chin, M., Dentener, F., Diehl, T., Easter, R., Fillmore, D., Ghan, S., Ginoux, P., Grini, A., Horowitz, L., Koch, D., Krol, M. C., Landing, W., Liu, X., Mahowald, N., Miller, R., Morcrette, J.-J., Myhre, G., Penner, J., Perlwitz, J., Stier, P., Takemura, T., and Zender, C. S.: Global dust model intercomparison in AeroCom phase I, Atmos. Chem. Phys., 11, 7781–7816, doi:10.5194/acp-11-7781-2011, 2011.

Kaiser, J. C., Hendricks, J., Righi, M., Jöckel, P., Tost, H., Kandler, K., Weinzierl, B., Sauer, D., Heimerl, K., Schwarz, J. P., Perring, A. E. and Popp, T.: Global aerosol modeling with MADE3(v3.0) in EMAC (basedonv2.53): model description and evaluation, Geosci. Model Dev., 12, 541–579, doi:10.5194/gmd-12-541-2019, 2019.

Kirkevåg, A., Iversen, T., Seland, Ø., Hoose, C., Kristjánsson, J. E., Struthers, H., Ekman, A. M. L., Ghan, S., Griesfeller, J., Nilsson, E. D., and Schulz, M.: Aerosol–climate interactions in the Norwegian Earth System Model – NorESM1-M, Geosci. Model Dev., 6, 207–244, doi:10.5194/gmd-6-207-2013, 2013.

Kirkevåg, A., Girni, A., Olivié, D., Seland, Ø., Alterskjær, K., Hummel, M., Karset, I. H. H., Lewinschal, A., Liu, X., Makkonen, R., Bethke, I., Griesfeller, J., Schulz, M. and Iversen, T.: A production-tagged aerosol module for Earth system models,OsloAero5.3 – extensions and updates for CAM5.3-Oslo, Geosci. Model Dev., 11, 3945-3982, doi: 10.5194/gmd-11-3945-2018, 2018.

Koch, D., Schulz, M., Kinne, S., McNaughton, C., Spackman, J. R., Balkanski, Y., Bauer, S., Berntsen, T., Bond, T. C., Boucher, O., Chin, M., Clarke, A., De Luca, N., Dentener, F., Diehl, T., Dubovik, O., Easter, R., Fahey, D. W., Feichter, J., Fillmore, D., Freitag, S., Ghan, S., Ginoux, P., Gong, S., Horowitz, L., Iversen, T., Kirkevåg, A., Klimont, Z., Kondo, Y., Krol, M., Liu, X., Miller, R., Montanaro, V., Moteki, N., Myhre, G., Penner, J. E., Perlwitz, J., Pitari, G., Reddy, S., Sahu, L., Sakamoto, H., Schuster, G., Schwarz, J. P., Seland, Ø., Stier, P., Takegawa, N., Takemura, T., Textor, C., van Aardenne, J. A., and Zhao, Y.: Evaluation of black carbon estimations in global aerosol models, Atmos. Chem. Phys., 9, 9001–9026, doi:10.5194/acp-9-9001-2009, 2009.

Koffi, B., and 32 coauthors: Evaluation of the aerosol vertical distribution in global aerosol models through comparison against CALIOP measurements: AeroCom phase II results, J. Geophys. Res. Atmos., 121, 7245-7283, doi:10.1002/2015JD0024639, 2015.

Lauer, A., Eyring, V., Hendricks, J., Jöckel, P. and Lohmann, U.: Global model simulations of the impact of ocean-going ships on aerosols, clouds, and the radiation budget, Atmos. Chem. Phys., 7, 5061-5079, doi:10.5194/acp-7-5061-2007, 2007.

Lee, Y. H., Adams, P. J., and Shindell, D. T.: Evaluation of the global aerosol microphysical ModelE2-TOMAS model against satellite and ground-based observations, Geosci. Model Dev., 8, 631–667, doi:10.5194/gmd-8-631-2015, 2015.

Mann, G. W., Carslaw, K. S., Spracklen, D. V., Ridley, D. A., Manktelow, P. T., Chipperfield, M. P., Pickering, S. J. and Johnson, C. E.: Description and evaluation of GLOMAP-mode: a model global aerosol microphysics model for the UKCA composition-climate model, Geosci. Model Dev., 3, 519-551, doi:10.5194/gmd-3-519-2010, 2010.

Mann, G. W. and 51 coauthors: Intercomparison and evaluation of global aerosol microphysical properties among AeroCom models of a range of complexity, Atmos. Chem. Phys., 14, 4679-4713, doi:10.5194/acp-14-4679-2014, 2014.

Michou, M., Nabat, P., and Saint-Martin, D.: Development and basic evaluation of a prognostic aerosol scheme (v1) in the CNRM Climate Model CNRM-CM6, Geosci. Model Dev., 8, 501–531, doi:10.5194/gmd-8-501-2015, 2015.

Pringle, K. J., Tost, H., Message, S., Steil, B., Giannadaki, D., Nenes, A., Fountoukis, C., Stier, P., Vignati, E. and Lelieveld, J.: Description and evaluation of GMXe: a new aerosol submodel for global simulations (v1), Geosci. Model Dev. 3, 391-412, doi:10.5194/gmd-3-391-2010, 2010.

Righi, M., Hendricks, J. and Sausen, R.: The global impact of the transport sectors on atmospheric aerosol: simulations for year 2000 emissions, Atmos. Chem. Phys., 13, 9939-9970, doi:10.5194/acp-13-9939-2013, 2013.

Roy, S. G. and Chakrabarti, A.: Chapter 11 – A novel graph clustering algorithm based on discrete-time quantum random walk, Quantum Inspired Computational Intelligence, Research and Applications, 361-389, doi: 10.1016/B978-0-12-804409-4.00011-5, 2017.

Rousseeuw, P. J.: Silhouettes: a graphical aid to the interpretation and validation of cluster analysis, Comput. Appl. Math., 20, 53-65, doi:10.1016/0377-0427(87)90125-7, 1987.
Sugar, G. A. and James, G. M.: Finding the number of clusters in a Dataset, J. Am. Stat. Assoc., 98, 750-763, doi:  10.1198/016214503000000666, 2011.

Textor, C., Schulz, M., Guibert, S., Kinne, S., Balkanski, Y., Bauer, S., Berntsen, T., Berglen, T., Boucher, O., Chin, M., Dentener, F., Diehl, T., Easter, R., Feichter, H., Fillmore, D., Ghan, S., Ginoux, P., Gong, S., Grini, A., Hendricks, J., Horowitz, L., Huang, P., Isaksen, I., Iversen, I., Kloster, S., Koch, D., Kirkevåg, A., Kristjansson, J. E., Krol, M., Lauer, A., Lamarque, J. F., Liu, X., Montanaro, V., Myhre, G., Penner, J., Pitari, G., Reddy, S., Seland, Ø., Stier, P., Takemura, T., and Tie, X.: Analysis and quantification of the diversities of aerosol life cycles within AeroCom, Atmos. Chem. Phys., 6, 1777–1813, doi:10.5194/acp-6-1777-2006, 2006.